# Controllability and Robustness of Functional and Structural Connectomic Networks in Glioma Patients

**DOI:** 10.3390/cancers15102714

**Published:** 2023-05-11

**Authors:** Anke Meyer-Baese, Kerstin Jütten, Uwe Meyer-Baese, Ali Moradi Amani, Hagen Malberg, Andreas Stadlbauer, Thomas Kinfe, Chuh-Hyoun Na

**Affiliations:** 1Department of Scientific Computing, Florida State University, Tallahassee, FL 32306, USA; ameyerbaese@fsu.edu; 2Institute for Biomedical Engineering, Technical University of Dresden, 01069 Dresden, Germany; 3Department of Neurosurgery, RWTH Aachen University, 52074 Aachen, Germany; 4Department of Electrical and Computer Engineering, Florida State University, Tallahassee, FL 32310, USA; 5School of Engineering, RMIT University, Melbourne, Victoria 3001, Australia; 6Department of Neurosurgery, Friedrich-Alexander University of Erlangen-Nürnberg, 91054 Erlangen, Germany; 7Center for Integrated Oncology Aachen Bonn Cologne Duesseldorf (CIO ABCD), 52074 Aachen, Germany

**Keywords:** glioma, driver set, structural and functional connectomics, critical nodes, controllability, robustness

## Abstract

**Simple Summary:**

Gliomas are known to impact on large-scale networks beyond the tumor location, but it is unknown how the tumor affects controllability and robustness of neural networks. We applied advanced control theory algorithms on connectivity data of structural and functional networks of prognostically differing glioma patients and healthy controls. We determined the driver nodes of the default-mode network (DMN), which are receptive to outside signals, and critical nodes as the most important elements for network controllability. Patients showed decreased network controllability and robustness mainly in the isocitratedehydrogenase (IDH) wildtype group, while additional topological shifts of driver and critical nodes were observed mainly in the prognostically more favourable IDH mutated patients. We hereby suggest a novel approach for elucidating disease evolution in brain cancer, which may aid in defining potential treatment targets under the aspects of network controllability and robustness in glioma patients.

**Abstract:**

Previous studies suggest that the topological properties of structural and functional neural networks in glioma patients are altered beyond the tumor location. These alterations are due to the dynamic interactions with large-scale neural circuits. Understanding and describing these interactions may be an important step towards deciphering glioma disease evolution. In this study, we analyze structural and functional brain networks in terms of determining the correlation between network robustness and topological features regarding the default-mode network (DMN), comparing prognostically differing patient groups to healthy controls. We determine the driver nodes of these networks, which are receptive to outside signals, and the critical nodes as the most important elements for controllability since their removal will dramatically affect network controllability. Our results suggest that network controllability and robustness of the DMN is decreased in glioma patients. We found losses of driver and critical nodes in patients, especially in the prognostically less favorable IDH wildtype (IDHwt) patients, which might reflect lesion-induced network disintegration. On the other hand, topological shifts of driver and critical nodes, and even increases in the number of critical nodes, were observed mainly in IDH mutated (IDHmut) patients, which might relate to varying degrees of network plasticity accompanying the chronic disease course in some of the patients, depending on tumor growth dynamics. We hereby implement a novel approach for further exploring disease evolution in brain cancer under the aspects of neural network controllability and robustness in glioma patients.

## 1. Introduction

Diffusely infiltrating glioma impacts large scale neural circuits [1,2,3,4] and the structural connectivity of the brain [5,6,7,8] beyond the apparent lesion site. Systemic functional connectivity (FC) changes have previously been found in glioma patients using resting-state functional MRI (rs-fMRI) and have been reported to differ depending on tumor growth dynamics, with stronger FC alterations found in high-grade compared to low-grade glioma patients [1,9]. A recent study, furthermore, described the increased efficiency of surrounding functional networks in lower grade glioma, while high-grade glioma mainly showed disruptions in brain functional connectivity in remote areas [2]. Likewise, structural disintegration with degradation of white-matter structures and alterations in the systemic microarchitecture of the brain have been reported as well [7,8,10], using diffusion MRI to analyze the microstructural integrity or estimate the number of streamlines connecting specific brain regions with each other [11,12]. The disease has a chronically progressive nature, with network alterations undergoing constant, dynamic changes, tumor-induced structural degradation, network disintegration, and diachisis on one hand, but potentially adaptive alterations and neuroplasticity (more likely to occur in slowly growing tumor types) on the other hand [2,13,14,15,16,17]. If, however, tumor-induced structural decline and network disintegration override the capacity for functional compensation, network failure and functional decline inevitably ensues. The dynamic interaction between the tumor and the structural and functional connectome and the relation between structural and functional connectivity alterations are, however, not sufficiently understood.

Modern dynamic graph network theory techniques and control theory applied to neural networks open a new research avenue which may aid in better understanding the dynamic properties of the structural and functional connectome in glioma patients. In the brain, we consider neural ensembles or regions as nodes of a graph. These are connected by edges which represent anatomical wires in a complex graph architecture, impacting disease and rehabilitation, besides neural function and development. It has been shown [18,19] that node-removals are more damaging than edge-removals to the network controllability, and that heterogeneous networks are more at risk than homogeneous networks. We apply advanced control theory to show how structural features of the graph network can inform temporal features of disease dynamics. The trajectory of this system represents the temporal path that the system undergoes through different states. A state is defined as the magnitude of neurophysiological and pathological changes across brain regions at a single point in time. Controllability of the graph network refers to the possibility to move the system along a desired trajectory. This means the system traverses a number of states from an initial one to a target state. We postulate that particular brain regions or nodes in the graph network at critical locations within the anatomical framework play the role of drivers that direct the system into specific modes of action. In clinical applications, it is essential to determine whether the brain as dynamic system can be controlled for applying therapeutic solutions. Consequently, network controllability becomes a focal research topic in many neurobiological applications. Regarding neurodegenerative diseases [20,21,22], it has previously been shown that certain brain regions or nodes in the connectivity graph can act as drivers for connectomic networks associated with Alzheimer’s disease and Mild Cognitive Impairment, and that these influence cognitive functions. Data on network controllability in tumor patients is, however, lacking.

The computational tools derived from control theory may allow us to address fundamental gaps in our understanding of disease evolution relevant to glioma. Looking into brain networks’ graph architectures, we discover highly connected and weakly connected areas. Two types of nodes are of particular interest in this context: the driver and the critical nodes. The driver nodes are the nodes which can send signals to target nodes, and thus are able to control the disease evolution trajectory. In a reverse engineering framework for glioma, these target points might be destinations for therapeutic intervention. A node is considered critical if its removal increases the number of drivers needed to maintain controllability. The critical nodes thus represent the brain regions that might be essential for preserving controllability of specific network functions, and which thus should be preserved in therapeutic intervention. 

We therefore developed and applied a control theory framework revisiting a connectomic data set of a previously published study [23] with prognostically differing glioma types (IDH mutated and IDH wildtype gliomas) [24], in order to identify nodes of SC and FC networks that can act as drivers and move the system (brain) into specific states of action. The driver nodes determined in the glioma graph networks change over time since the brain network structure and connectivity undergoes alterations in the course of the disease. Pinning controllability techniques provides us with the set of those driver nodes. Brain network failures are associated with dysfunction and disease progression. The controllability of a network decreases as a result of such failures, which is reflected by an increase in the number of drivers needed to control the network. To resist failures, strong robustness is preferable, and often required, in the context of biological networks. Therefore, we consider the correlation between brain network robustness and topological features extremely important in the context of understanding disease evolution, and hereby suggest a novel approach for further exploring disease evolution in brain cancer under the aspects of controllability and robustness of structural and functional neural networks.

## 2. Materials and Methods

### 2.1. Participants

In the initial study, 29 glioma patients (mean age: 50 years, 17 males, 28 right-handed, 19 left-hemispheric tumors) and 27 healthy controls (mean age: 46 years, 17 males, 26 right-handed) were included [25]. Only patients between 18 and 80 years with unilateral supratentorial tumors and a Karnofsky index of ≥70 were included in the study. Patients were enrolled preoperatively and all except two of the patients were naïve to prior tumor-specific treatment. Histopathological diagnoses were determined according to the revised WHO tumor classification of 2016 [26], integrating histoanatomical and moleculargenetic criteria under consideration of the IDH-mutation status and codeletion of chromosome arms 1p and 19q in each patient. Patients’ demographics and tumor characteristics can be found in Table 1. Patients were split into subgroups according to their IDH-mutation status, resulting in patient groups of 15 IDHmut and (mean age: 37 years, 11 males, 14 right-handed, 10 left-hemispheric tumors) and 14 IDHwt patients (mean age: 65 years, 6 males, 14 right-handed, 9 left-hemispheric tumors). Tumor volumes did not significantly differ (*p* < 0.062) between IDHmut and IDHwt patients. 

### 2.2. MRI Data Acquisition

For the present study, an MRI data set previously obtained and published [23] was used for the analyses. The detailed scanning protocol is therefore described in detail elsewhere [3,10,23] and comprised the following pulse sequences: a sagittal 3D T1 magnetization-prepared rapid acquisition gradient echo (MPRAGE) sequence; a contrast-enhanced, T1-weighted turbo inversion recovery magnitude (TIRM) dark-fluid sequence; a T2-weighted TIRM dark-fluid scan; a fluid attenuation inversion recovery (FLAIR); as well as an echo planar imaging (EPI) sequence, including 300 whole-brain functional volumes. Scanning was applied using a 3T Siemens Prisma MRI scanner equipped with a standard 20-channel head coil. (For a more detailed description please see Appendix A).

#### 2.2.1. Data Preprocessing

Preprocessing steps were applied as described in detail previously [23]. In brief, tumor masks were semi-automatically created using ITK-SNAP software (version 3.6.0, Paul A. Yushkevich, University of Pennsylvania, USA and included T1 hypo- and T2-FLAIR hyperintense lesions for gliomas grade II-III, as well as T1 hypointensities and contrast-enhancing lesions for glioblastomas. Tumor masks were manually corrected by an experienced neurosurgeon and used for cost-function masking in the preprocessing. Functional preprocessing included the realignment of functional images and their coregistration to the strucutral image, the normalization to MNI space, smoothing, and high-pass filtering of functional images. A whole-brain parcellation was then applied using the Brainnetome Atlas (https://atlas.brainnetome.org, accessed on 20 February 2020), generating 246 predefined anatomical brain regions that served as ROIs for the following analyses (for further details please see Appendix A).

#### 2.2.2. Data Analyses

Within these ROIs, SC and FC were computed by generating 246 × 246 whole-brain SC and FC connectivity matrices for each subject. Briefly, SC measures were based on probabilistic diffusion tractography and comprised edge-weight (EW), as determined by the number of fiber connections between two ROIs divided by the mean number of fibers originating or ending within these two ROIs. For FC, mean time-series were generated for corresponding ROIs and correlated (including a Fisher z transformation) with each other (for further details please see Appendix A). As a result, a 246 × 246 SC and FC connectivity matrix was generated for each subject, which were then averaged per group to obtain one mean SC and one mean FC matrix for each IDHmut, IDHwt, and the healthy control group. 

In this paper, we analyzed structural and functional connectivity matrices obtained in the previous study [23] under the aspect of network controllability and robustness of the DMN in IDHmut and IDHwt patients compared to healthy controls. In the current analyses, only DMN-related ROIs extracted from the Brainnetome Atlas were included, comprising the anterior cingulate cortex, posterior cingulate cortex, medial temporal gyrus, and inferior parietal lobule. From the original 188 × 188 correlation matrix set [23], we thus extracted a total of 26 non-zero rows/columns with 13 ROIs for the right hemisphere (12, 14, 42, 82, 84, 136, 138, 144, 152, 154, 176, 182, 188) and 13 ROIs for the left hemisphere (11, 13, 41, 81, 83, 135, 137, 143, 151, 153, 175, 181, 187), as labelled according to the Brainnetome atlas. The nodes in the resulting graphs keep the original ROIs’ numbers obtained according to the Brainnetome atlas, and we are consistent with the numbering throughout our processing (for a detailed list of included DMN ROIs please see Appendix A).

### 2.3. Methods for Generating Structural and Functional DMN Graphs

There are many metrics based on statistical dependencies used in defining brain connectivity, but the most common is correlation. Thus, we thresholded the sample connectivity matrices. Based on these 26 × 26 matrices the adjacency matrix and graphs were computed. We determined the driver of the graph by using the standard procedure of computing the column canonical form (CCF) and eigenvalues mode (i.e., most frequent value), as has been described in detail previously [27]. The drivers are identified by finding the linear dependent row vectors in the CCF. According to [27], we obtained a graph with N = 26 nodes for a chain network with one driver, for a star network we obtained N − 2 = 24 drivers, and for a fully connected network we obtained N − 1 = 25 drivers.

As there is no optimal threshold for generating structural and functional graphs, in general, a range of thresholds is considered. In this paper, we adopted from literature a range of common thresholding schemes [28] and applied them to DMN SC and FC connectivity values.

First, we implemented the lowest percentage threshold method by removing only 2.5%, 5%, 10%, 20%, and 30% of the edges. Increasing the threshold will decrease the number of drivers and produce a compressed information representation. For these lowest threshold methods, we always used the threshold computed from the healthy control group for the three sets (for each SC and FC).

In addition, we implemented three further pre-processing methods to generate graphs besides the original zero threshold method. These are the adaptive threshold method, the partial correlation method, and the Chow-Liu algorithm:

Adaptive Threshold Method (ATH): We implemented an adaptive threshold method by applying an iterative algorithm to determine the largest threshold value possible (with an error < 10^−4^) such that the adjacency matrix still produces a fully connected graph. This graph has a reduced number of edges compared to the original data but still has essentially more edges than the CLA method. This will generate six different thresholds for the six data sets.

Partial Correlation Method (PCM): The partial correlation method assumes that the data correlation matrix is positive and definite. The inverse of the data matrix is determined and normalized afterwards. To obtain a sparse matrix, we apply a soft threshold since we have a finite number of samples and noise.

Chow-Liu Tree Algorithm (CLA): This method implements the maximum weight spanning tree of the connectivity graph [25,29]. Unlike PCM, it does not require a matrix inversion and can be directly applied to the original data matrix. The method constructs the optimal tree but will produce, in case the graph is disconnected, false positives. Figure 1 summarizes the employed processing steps.

### 2.4. Network Controllability and Driver Nodes

A network can be described by a graph *G* = (*V*, *E*), where *V* = {1, 2, …, *N*} is the set of nodes and $E=\{e_{ij}|i,j\in V,i\neq j$} is the set of *M* links. In this paper, we consider undirected networks. In [4], it was shown that a simple noise-free linear discrete-time and time-invariant model can be employed to describe the neural dynamics measured by fMRI
(1)$$x\left(t+1\right)=Ax\left(t\right)+B_{K}u_{K}\left(t\right)$$
where $x\in R^{n}$ is the magnitude of the neurophysiological activity (state) of the brain regions over time, $A\in R^{n\times n}$ is the adjacency matrix, $B_{K}\in R^{n\times m}$ is the input matrix, and $u_{K}\in R^{m}$ is the input vector. The input matrix identifies the control nodes, *K*, in the brain. The interaction network is presented by its Laplacian matrix, *L* = [*l*_ij_] where *l_ij_* = −1; *i* ≠ *j*, if there is a link between nodes *i* and *j*, and 0 otherwise [30]. The *i^th^* diagonal element is $l_{ii}=-\sum_{j}l_{ij}$, which forms the Laplacian as a zero-row-sum matrix. 

### 2.5. Robustness Measures

The dynamic behavior of a brain network is strongly linked to the topological structure of the network represented by nodes and edges. Changes in network parameters would affect both the robustness and stability of the network. Robustness is an important feature of brain networks because it describes the ability of the network to resist random or targeted attacks. We view the glioma development as an attack on the connectivity of the brain networks. We employed two measures here to characterize robustness based on the eigenvalues of the Laplacian or adjacent matrix: the algebraic connectivity and the natural connectivity.

Algebraic connectivity: The algebraic connectivity *α*(*G*) is defined as the second smallest eigenvalue of the Laplacian matrix.
(2)$$\alpha \left(G\right)=\lambda_{2},\lambda_{1}\leq \lambda_{2}\leq \lambda_{3}\leq \cdots {\leq \lambda }_{N},$$
where $\lambda_{i},i=1,2,3,\ldots ,N$ are the eigenvalue of Laplacian matrix of graph *G*.

Natural connectivity: Natural connectivity is defined as an average eigenvalue that changes strictly monotonically with the addition or removal of edges.
(3)$$\overset{-}{\lambda }=ln(\frac{1}{N}\sum_{i=1}^{N}e^{\lambda_{i}})$$
where $\lambda_{i}$ is the eigenvalue of the adjacency matrix. An empty graph has the minimum natural connectivity while the complete graph has maximum connectivity. A larger value is associated with a more robust graph.

### 2.6. Network Controllability under Vulnerability: Determination of Critical Nodes

Critical nodes represent a subset of nodes in the network whose removal leads to a decline in performance metrics such as algebraic and natural connectivity. These attacks lead to a reduced topological stability and reflect an altered cerebral organization. To determine the critical nodes, we remove a node of the network and recompute the number of driver nodes. If the number goes up, we identify the removed node as a critical node (please see Figure 1).

## 3. Results

### 3.1. Implementation of Pre-Processing and Graph-Generating Methods

The lowest percentage threshold method showed, at a threshold of 30%, the most reliable results and discriminated best between healthy controls and glioma patients.

The adaptive threshold method (ATH) determined the largest threshold, such that a fully connected graph was produced. The number of edges was reduced compared to the zero-threshold method but yielded more than the CLA. We thus obtained six different thresholds for the six data sets. Applying ATH, the drivers had a range of 1–4 (mean = 1.875; median = 1 across the 6 data sets) out of 26 nodes.

The partial correlation method (PCM) generated the inverse of the correlation matrix and normalized the output results. In [28], thresholding for medical imaging application was recommended. The number of drivers was now in the range 1–15 with a mean = 4.375 and median = 2. A DMN/IDMmut/IDHwt classification trend was not observed by using PCM.

The Chow-Liu tree algorithm (CLA) reduced the number of edges, N – 1, to the minimum possible, such that the graph was fully connected. For N = 26 nodes, this always gives a graph with 25 edges. CLA tries to identify the graph with maximum weight by first sorting all edge weight and then adding only the weights and nodes that are not corrected. The number of drivers was low (range 0–6; median = 2; mean = 2.78). Regarding the DMN/IDMmut/IDHwt classification, no clear trend was observed, as the number of drivers was often too low due to the few edges in the graph. The CLA method generated a fully connected graph but, by the definition of the algorithm, many large correlation values were removed from a graph if the edge already had a connection. Therefore, important information was lost from the graph, which seemed counterproductive in a medical setting, so that the CLA method appeared to be not suitable in the present context.

### 3.2. Driver Nodes of Structural and Functional Graphs

We therefore applied the lowest percentage threshold method for determination of the driver nodes of structural and functional networks in patients and controls. Very often the same threshold is used across different groups for comparison [31]. Therefore, the first five threshold values to eliminate 2.5%, 5%, 10%, 20%, and 30% of the connectivity values/edges were computed always using the DMN sets. Here, 2.5% would be equivalent to the *µ* − 2*a*-threshold of a Gaussian distribution. The upper limit for edge removal was 38% to produce still a fully connected graph. The 26 × 26 matrix data were sorted, percentage values to compute the threshold were determined, and then the threshold was applied.

Threshold values were first computed for the control group, both for the SC and FC DMN graph, and these thresholds were then also applied to compute the graphs and drivers for the patient data (IDMmut and IDHwt). A substantially clearer trend was now visible in terms of the number of drivers: the number in the control group was, in the majority of cases, higher than the number in the IDMmut and IDHwt graphs, as can be seen Figure 2.

We then determined the driver nodes for structural and functional networks for controls and glioma cases as an average computed over the five thresholds using the lowest percentage threshold method (Table 2).

We show, as an example, the location of driver nodes at a 30% threshold (TH). 

In Figure 3, the nine driver nodes for the structural DMN in healthy controls are located in the frontal lobe (orbital gyrus), parietal lobe (inferior parietal lobule, precuneus, postcentral gyrus), and the limbic lobe (cingulate gyrus). For IDHmut, five driver nodes are located in the parietal lobe (inferior parietal lobule) and the limbic lobe (cingulate gyrus). For IDHwt, four driver nodes were found in the parietal lobe (inferior parietal lobule, precuneus) and the limbic lobe (cingulate gyrus). 

In Figure 4, the four driver nodes for the functional DMN in healthy controls are located in the precuneus and in the cingulate gyrus. In IDHmut patients, three driver nodes are located in the orbital gyrus, middle temporal gyrus, and in the cingulate gyrus. For IDHwt patients, only one driver node was found and located in the cingulate gyrus. 

We analyzed these results for SC and FC and observed the following:The number of driver nodes (DN) for both SC and FC was decreased in patients compared to controls, with the lowest number in the prognostically less favorable IDHwt group (see Figure 3 and Figure 4).IDHmut patients showed not only losses, but also topological shifts with “alternative” DN compared to controls for SC in the cingulate cortex (left dorsal area 23, left subgenual area 32) and for FC in the right orbital gyrus and left middle temporal gyrus (see Figure 5).Common DN for both patients and controls were found in the inferior parietal lobe (IPL) and in the cingulate cortex (left ventral area 23, right subgenual area 32) for SC, and in the left cingulate cortex (subgenual area 32) for FC.Healthy controls showed common DN for SC and FC in the cingulate cortex (right ventral area 23), IDHmut patients showed common DN for SC and FC in the cingulate cortex and in left subgenual area 32, while IDHwt showed no common DN across SC and FC networks.

In summary, patients and controls share common drivers for SC in the cingulate cortex and the right inferior parietal lobe and, for FC, in the cingulate cortex. While the absolute number of driver nodes decreased in patients, losses are higher in the prognostically less favorable IDHwt group, with topological shifts found in the IDHmut but not in the IDHwt group.

### 3.3. Robustness of Structural and Functional DMN

The algebraic connectivity is determined by the second smallest eigenvalue of the Laplacian matrix. It was shown by [32] that the magnitude of this measure reflects how well connected the graph is.

Figure 6 shows the values of the robustness measure algebraic connectivity for the SC and FC networks. The SC networks for the unthresholded connectivity graph show a decreased robustness for glioma patients compared to controls. The lowest robustness is achieved for the less favorable IDHwt. This confirms that the reduced topological stability is due to altered cerebral organization. The other methods showed less pronounced differences due to the fact that the number of edges varied less within each group of graphs. The FC networks also showed a less pronounced behavior regarding robustness compared to the SC networks. Their variations were minimal between control and glioma networks for almost all methods.

Since the algebraic connectivity is too coarse to describe the structural connectivity, the natural connectivity was proposed by [33] as an alternative. It represents an average eigenvalue that changes monotonically when edges are deleted or added. The larger the value of natural connectivity, the more robust is the graph. The results in Figure 7 show the same trend as those in Figure 6 for the SC networks. For the FC networks for ATH and PCM, we observed a decrease in robustness of the IDHwt network compared to controls and IDHmut. 

### 3.4. Topological Vulnerability for Structural and Functional Networks

We investigated the robustness for both healthy controls and glioma patients and determined the critical nodes, observing differences between groups (Table 3).

We analyzed the results from Table 3 for SC and FC and observed the following: (a)The number of critical nodes (CN) for FC but not SC decreased in patients compared to controls, with the lowest number in the prognostically less favorable IDHwt group (see Figure 8 and Figure 9).(b)Patients showed not only losses, but also additional CN compared to controls for SC and FC, with more “alternative” CN being encountered in IDHmut than in IDHwt patients (please see Figure 10).(c)Common CN for patients and controls were found in the inferior parietal lobe (IPL, rostroventral area 39) and middle temporal gyrus (MTG) regarding SC. Common CN for patients and controls regarding FC were found in the superior frontal gyrus (SFG), inferior parietal lobe (IPL, caudal area 39), and the precuneus. (d)Healthy controls showed common CN for SC and FC in the right and left MTG (rostral area 21). IDHmut patients had common CN in the right and left MTG (rostral area 21), and in the right IPL. IDHwt patients showed common CN for SC and FC only in the right MTG (caudal area 22).

In summary, patients and controls shared common critical nodes in the MTG and IPL for SC, and in the SFG, IPL and precuneus for FC. The number of critical nodes decreased in patients regarding FC, especially in the IDHwt group. The number of CN regarding SC remained stable or even increased in the IDHmut group, mainly due to additional recruitment of “alternative” CN compared to controls, with more topological shifts being observed in the IDHmut than in the IDHwt patient group. 

Notably, driver and critical nodes were differently distributed both within structural and functional networks.

## 4. Discussion

The aim of the present study was to investigate the controllability of neural networks in glioma patients compared to healthy controls by applying control theory algorithms on functional and structural connectomic MRI data. To the best of our knowledge, this is the first study to determine driver as well as critical nodes in a real neural system in the context of brain cancer. Regarding the DMN, we found a loss in driver (DN) and critical nodes (CN) in tumor patients, which appeared to be even more pronounced in the prognostically less favorable IDHwt group. However, not only losses, but also topological shifts were observed for DN as well as for CN. This applied for structural as well as functional DMN-connectivity, with a higher number of “alternative” DN and CN being found in IDHmut compared to the IDHwt patient group. Robustness was decreased in patients compared to controls, which again appeared to be more pronounced in IDHwt patients, mainly with regard to the structural DMN. Thus, network controllability was decreased in glioma patients, and seemed to differ in prognostically differing tumor types. While the losses of DN and CN may reflect tumor-induced disintegration of structural and functional neural circuits, the topological shifts and recruitment of alternative nodes might be related to neuroplasticity accompanying the chronic disease course, which is more likely to occur in the slower growing lesions, and which applies to the prognostically more favorable IDHmut patients in our study.

An increasing number of studies [34,35] describe associations of intrinsic network features with glioma predilection sites and potential tumor growth trajectories. While recent studies suggest preexisting network characteristics impact tumor localizations and tumor growth, there are also indications that premorbid network features do not remain unaltered by the tumor pathology but undergo constant tumor-induced changes such as network degradation and plasticity accompanying the disease course [13,15,16,17]. Therefore, it has to be assumed that at the time of diagnosis, alterations of intrinsic network features may well have preceded disease manifestation, so that not only premorbid network characteristics will define lesion growth and functional sequelae, but also more complex and constant interactions with dynamic network alterations along the disease course. In that regard, our study complements previous findings by adding the new dimension of deciphering tumor effects on the inner functional network architecture from the perspective of network controllability, which may differ not only depending on tumor location but also on tumor growth dynamics. Therefore, observed differences in network controllability of prognostically differing glioma types may add to a better understanding of the dynamic interaction of different tumor lesion types with the functional and structural connectome.

### 4.1. Controllability of the DMN

We investigated controllability specifically of the DMN as the fundamental intrinsic network, as it is considered essential for human cognition and for the integration of different neural network functions. With the DMN showing anticorrelations to extrinsic networks during task performance and rest, the strength of this dichotomy has previously been linked to the level of task performance, and alterations of DMN controllability therefore may greatly impact on various other network functions. Considering that early functional imaging studies have defined the DMN with the ambiguity of being activated mainly in the resting condition while being deactivated during most active tasks, the DMN seems to be especially suited for investigating functional network controllability based on resting-state functional MRI acquisitions. Moreover, the DMN has been shown to be inter- and intraindividually highly robust, allowing for cross-sectional studies across different groups. As the DMN, the most commonly investigated intrinsic network, has been anatomically well defined, we chose to investigate DMN connectivity based on a whole-brain parcellation using the Brainnetome Atlas, selecting empirically defined DMN ROIs, which allow comparison between corresponding regions across different subjects. While tumor lesion masks were carefully used for the preprocessing procedures, tumor regions were not excluded from the analyses of functional and diffusion MRI data. Considering the infiltrative nature of glioma tumor cells along white matter bundles, tumor regions may still be functionally and/or structurally connected depending on tumor growth dynamics, so that the “network relevance” of a tumor lesion thus may not solely depend on tumor location or size, but also on the degree of local destruction on the one hand, and perilesional neuroplasticity accompanying the chronic disease course on the other hand. It was previously shown [1], that functional connectivity of glioma lesions significantly differed depending on the tumor grade. As core regions of the DMN encompass, in particular, cortical midline structures, the overlap with tumor regions was also limited. The differences observed here in network controllability across patient groups thus might also relate to different tumor growth dynamics, as indicated here by IDH mutation status. By restricting the analyses to predefined atlas-based regions of the DMN, we analyzed network alterations specifically within a well-defined anatomical framework. While the decrease in DN and CN may simply relate to tumor-induced network degradation, the here observed topological shifts of driver and critical nodes (in the case of critical nodes preserved or even increased in number in IDHmut patients), even within these well-defined anatomical boundaries, may indicate varying degrees of neuroplasticity in prognostically differing patient groups while not being able to account for network plasticity beyond this predefined anatomical framework with the current approach.

### 4.2. Network Controllability in Brain Tumor Patients

Random failures or attacks stemming from aberrant signals on brain networks are frequently happening and can have severe consequences. These failures or attacks are associated with node or edge removals, causing dysfunction or cancer in humans. Therefore, looking into network controllability robustness is of critical importance and it reflects how well brain networks can preserve their controllability against random failures or attacks. Attacks may either be random (e.g., a tumor pathology) or targeted (e.g., a therapeutic intervention). In [36], it was shown that node removals are more detrimental than edge removals to controllability; likewise, heterogeneous networks are more prone to connectivity destruction. In this sense, we determine the critical nodes [18] whose removal negatively impacts the network controllability. In general, the network controllability is supposed to decrease after attacks, which is shown by the increase in the number of drivers [36]. A node is defined as critical if, and only if, its removal increases the number of drivers needed to maintain the network controllability. In [18,36], it was shown that these critical nodes are extremely relevant for network controllability.

In the context of brain cancer, attacks on critical nodes result in the need for a higher number of DN in order to maintain network controllability, rendering the system less efficient and more vulnerable at the same time. In our study, CNs were not reduced in absolute numbers in patients due to the recruitment of alternative nodes. DN, on the contrary, were reduced in the patient group compared to controls, which may indicate the failure of the system to sufficiently compensate for the losses, and which might be paralleled by functional impairment in patients. Functional decline, e.g., in the cognitive domain, has frequently been described to be more pronounced in faster growing tumor lesions [37,38], irrespective of tumor volumes [9], which complies with the notion of network failure, especially in higher grade gliomas [1,2]. Characterizing network controllability in brain tumor patients might not only be relevant in terms of understanding functional sequelae but may also offer new means to investigate network vulnerability, e.g., regarding the propagation of epileptic seizures; to define treatment targets; and to improve prognostication of disease evolution. Regarding therapeutic interventions (e.g., surgical resection) as potential attack on network integrity, control theory algorithms may offer tools to identify those nodes (i.e., brain regions) which might be indispensable for maintaining controllability of specific network functions, and which thus should be preserved in therapeutic intervention. Assuming constant dynamic interactions between disease propagation on the one hand and attempts of compensation of the neural system on the other hand, using control theory algorithms for characterizing neural network dynamics may provide new means for adapting therapeutic strategies along the disease course: Repeated assessments of network controllability with (re-)evaluation of CN and DN topology may provide a better understanding of the “functional reserve capacity” of the network, and may allow adaptation of therapeutic targets along the disease’s course, given the topological shifts of DN and CN observed in our study in some of the patients. As such, applying control theory algorithms to neural networks might nurture the treatment concept of “prehabilitation”, referring to the possibility of adapting therapeutic targets along the disease course under special consideration and promotion of neuroplasticity preceding therapeutic (re-)intervention [39,40]. 

### 4.3. Limitations and Perspectives

We are, however, very well aware that the present study has some fundamental limitations. First and foremost, the small sample size constitutes a major limitation, in that other factors, such as tumor histology and other molecular genetic markers, tumor volume, and tumor location could not be investigated for their potential influences on network controllability. While we could not control for tumor location due to the small sample size, IDHmut and IDHwt at least did not differ significantly in tumor volumes in the present cohort. We furthermore cannot exclude that age differences across groups may have contributed to the observed differences in network controllability in our cohort, but the inherently differing age distributions of different glioma subtypes may generally not be easily overcome when comparing prognostically differing tumor types. Further methodological limitations are grounded in the imaging modalities applied, with diffusion MRI only providing gross estimates of white matter connectivity on a macroscopic scale, with known shortcomings especially related to fibre curvature and fibre crossings [41]. Considering the infinitely more complex cerebral structural organization with strong reciprocity and complex vertical as well as horizontal neural interconnections, as suggested by previous investigations on the microscale [42], the applied methods here can only aspire to provide a gross estimate and strongly simplified model of neural network controllability. Despite this, diffusion MRI still provides valuable insights into white matter microstructural architecture, which at present cannot be obtained by any other clinically available method in vivo. Analyzing functional connectivity based on resting-state functional MRI measures also has limitations, such as those caused by neurovascular uncoupling [43] or due to limited spatial and temporal resolution, but still provides valuable information on functional network architecture in a clinically feasible setting. 

We hereby introduced the concept of controllability and driver nodes that are recipients of outside signals (which are thus able to move the brain system into specific states of action) in the context of brain cancer. With regard to neurodegenerative diseases, it has previously been shown [22] that driver nodes are representative for the transitions between healthy controls, mild cognitive impairment, and Alzheimer’s. Application of control theory algorithms on neural network functions of brain tumor patients is, however, lacking. In a novel approach, we determined driver node locations in structural and functional brain networks and showed that their number and location is distinctive for healthy controls and glioma patients and point to changes in the brain network structure that may be attributed to the disease. Biological networks are robust against random failures but are at risk for targeted attacks on the so-called critical nodes. There is a close relationship between controllability and network robustness captured by the critical nodes and their significance in supporting robust brain functions. We believe that this study provides novel impulses and opens a new research avenue to determine brain network robustness at the nodal level, which is relevant for many brain diseases.

We claim that the information provided by controllability may be used in combination with other graph nodal measures for understanding global changes in glioma. An interesting future direction is to estimate the network connectedness robustness based on deep learning methods. Connectedness is an important and necessary property for ensuring controllability and is given by a sufficient number of properly connected nodes. Evaluating the importance of nodes is computationally expensive and AI methods help alleviating this problem. Connectedness robustness is more difficult to predict than controllability robustness since it has a higher variability than the latter.

## 5. Conclusions

In a novel approach, we analyzed SC and FC networks in glioma patients from the dynamic view of controllability and robustness of a neural system by determining the driver and critical nodes within the DMN. We observed losses in driver and critical nodes for both SC and FC in glioma patients compared to controls, indicating impaired network controllability and network robustness in patients, which was even more pronounced in the prognostically less favorable IDHwt patient group. While decrease in DN and CN may reflect tumor-induced disintegration of neural circuits, the topological shifts and recruitment of alternative nodes observed mainly in IDHmut patients might be related to neuroplasticity accompanying the chronic disease course, which is more likely to occur in slower growing tumor lesions. Applying cognitive control theories to structural and functional connectomic patient data opens new avenues for investigating neural network dynamics, which may aid in improving prognostication of disease evolution and functional outcome in therapeutic intervention. 

## Figures and Tables

**Figure 1 cancers-15-02714-f001:**
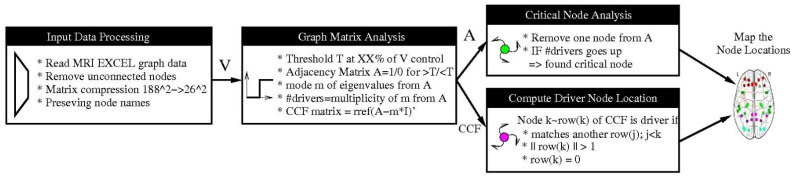
Number of drivers for the six data sets obtained by applying different threshold levels.

**Figure 2 cancers-15-02714-f002:**
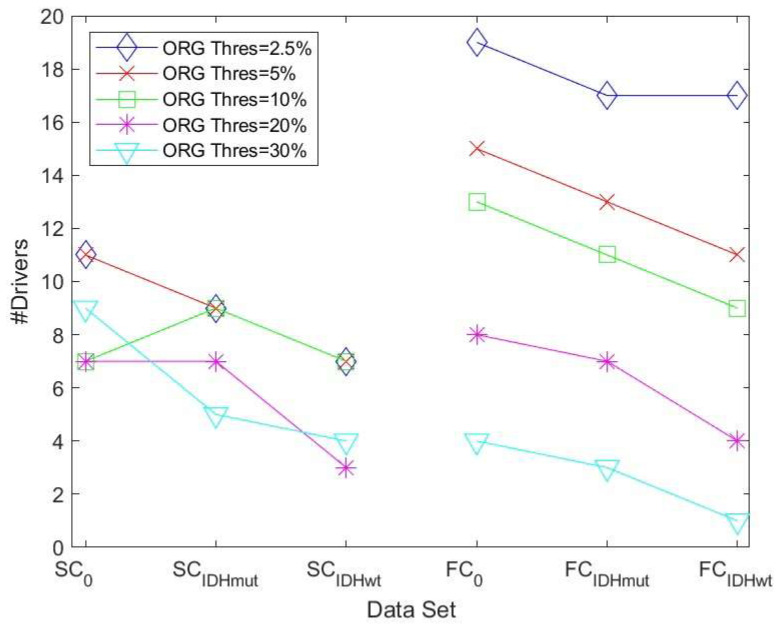
Number of drivers for the six data sets obtained by applying different thresholds using the lowest percentage threshold method. (SC refers to the structural DMN graph, FC to the functional DMN graph. 0 refers to the control group).

**Figure 3 cancers-15-02714-f003:**
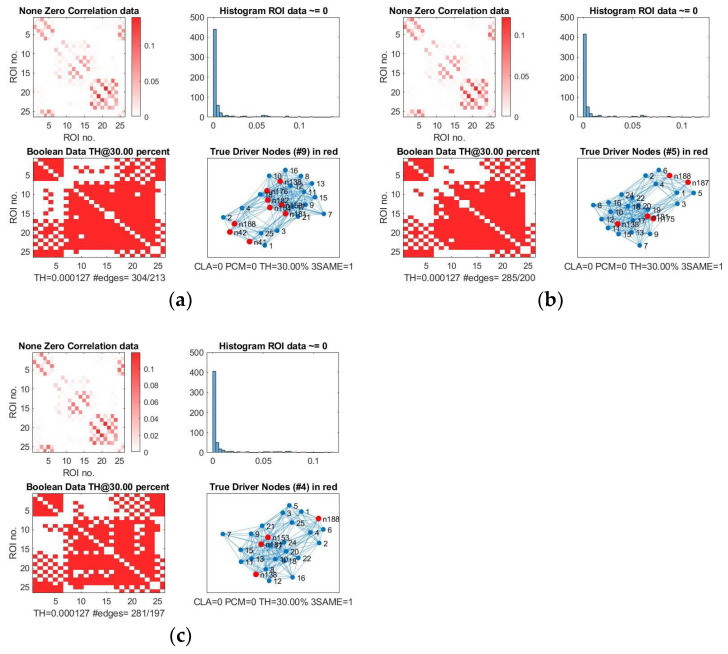
Driver nodes of the structural DMN (Edge Weight; EW). (**a**) Controls, (**b**) IDHmut, and (**c**) IDHwt subjects. In each of the three images we have: on the upper left, the original non-zero correlation data; on the upper right, the histogram of the correlation values found in the connectivity matrix; on the lower left, the Boolean data for the threshold at 30%; and on the lower right, the TH graph with the determined driver nodes. Driver nodes are in red.

**Figure 4 cancers-15-02714-f004:**
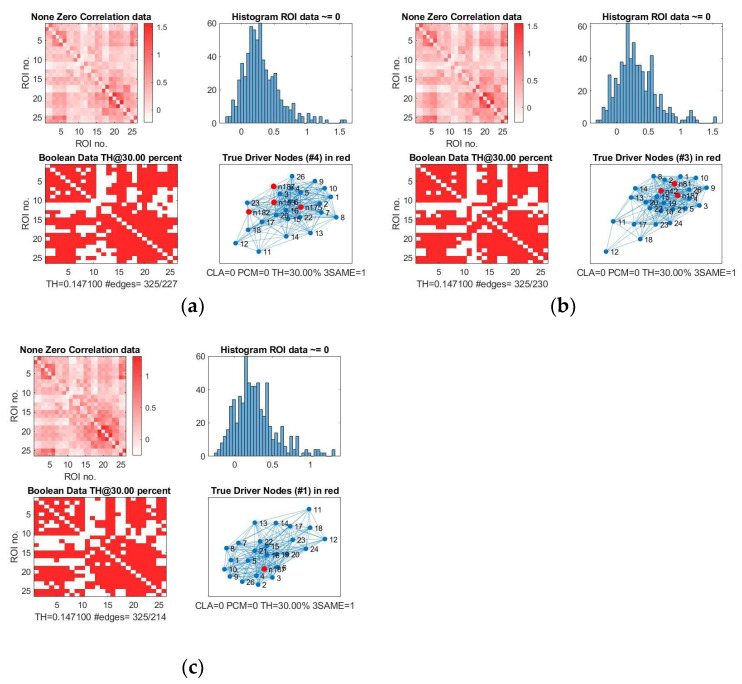
Driver nodes of the functional DMN, rsfMRI. (**a**) Controls, (**b**) IDHmut, and (**c**) IDHwt subjects. In each of the three images we have: on the upper left, the original non-zero correlation data; on the upper right, the histogram of the correlation values found in the connectivity matrix; on the lower left, the Boolean data for the threshold at 30%; and on the lower right, the TH graph with the determined driver nodes. Driver nodes are in red.

**Figure 5 cancers-15-02714-f005:**
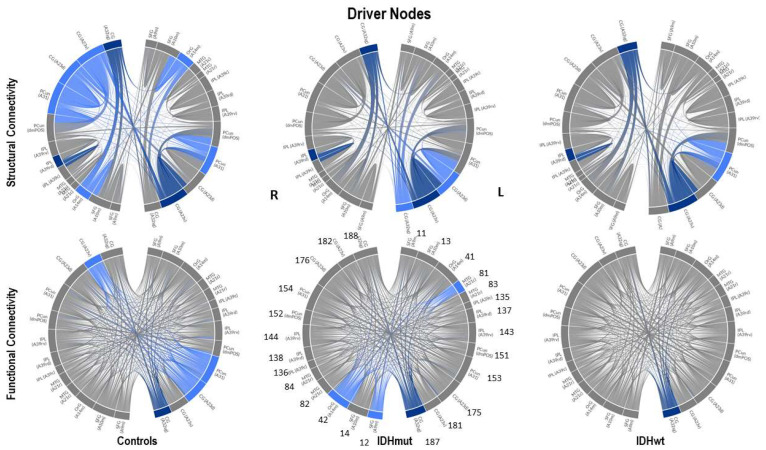
Connectograms for SC and FC for each group highlighting the driver node (DN) connections in blue. Given in dark blue are common DN across groups, both for SC and FC, respectively. DMN nodes are labeled numerically according to the ROIs as given in the Brainnetome Atlas (https://atlas.brainnetome.org, accessed on 20 February 2020). The total number of DN decreases in patients, with the lowest number in the IDHwt group. Only in the IDHmut patients, not only losses but also “alternative” DN are found compared to controls, indicating a topological shift in some of the DN.

**Figure 6 cancers-15-02714-f006:**
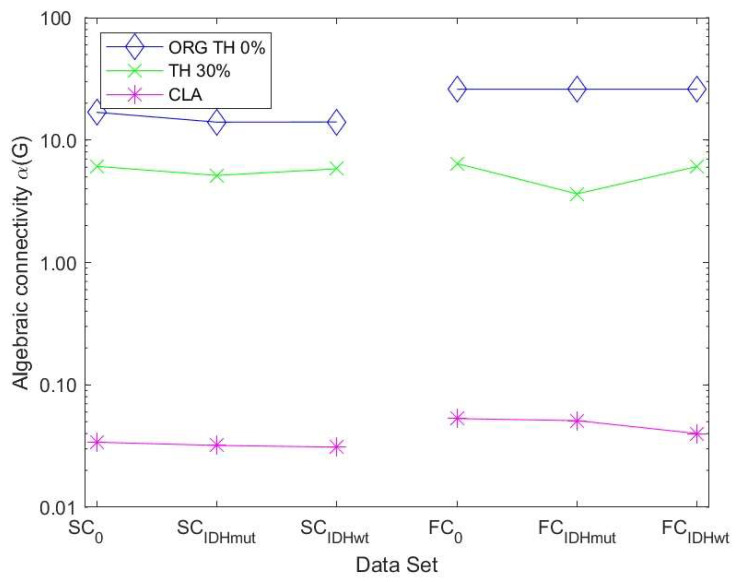
Algebraic connectivity values for SC and FC networks with examples for different thresholding methods: ORG TH 0% represents the unthresholded connectivity graph, TH 30% represents the thresholded graph at 30%, and CLA represents the graph built based on the Chow-Liu-algorithm. (SC refers to the structural DMN graph, FC to the functional DMN graph. 0 refers to the control group).

**Figure 7 cancers-15-02714-f007:**
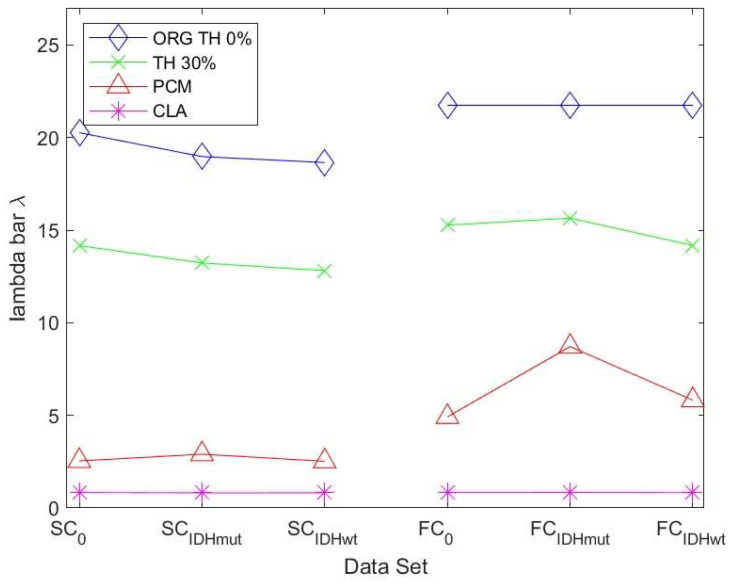
Natural connectivity for SC and FC networks with examples for different thresholding methods: ORG TH 0% represents the unthresholded connectivity graph, TH 30% represents the thresholded graph at 30%, PCM represents the partial correlation graph, and CLA represents the graph built based on the Chow-Liu-algorithm. (SC refers to the structural DMN graph, FC to the functional DMN graph. 0 refers to the control group).

**Figure 8 cancers-15-02714-f008:**
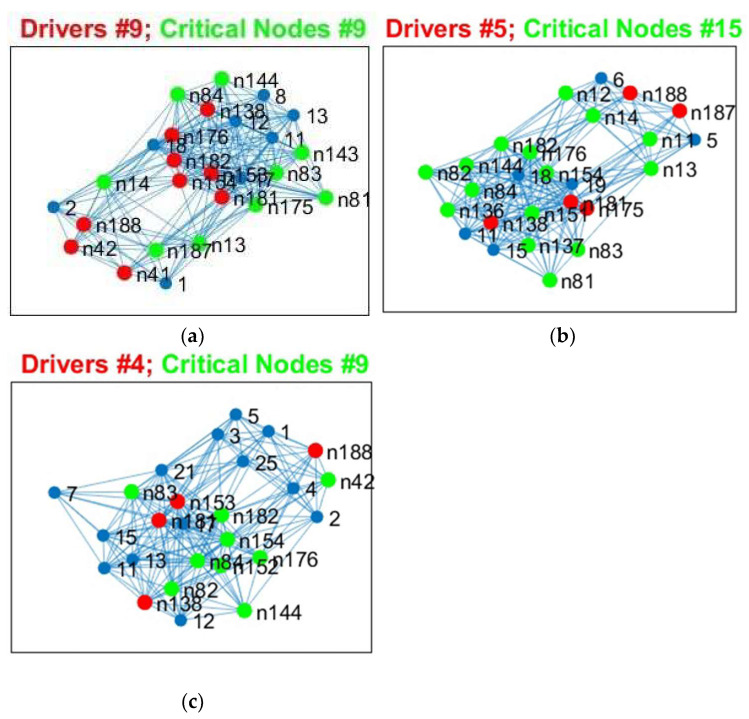
Critical Nodes for structural data, edge weight (EW). (**a**) Controls, (**b**) IDHmut, and (**c**) IDHwt subjects. Critical nodes are in green and initial driver nodes are in red.

**Figure 9 cancers-15-02714-f009:**
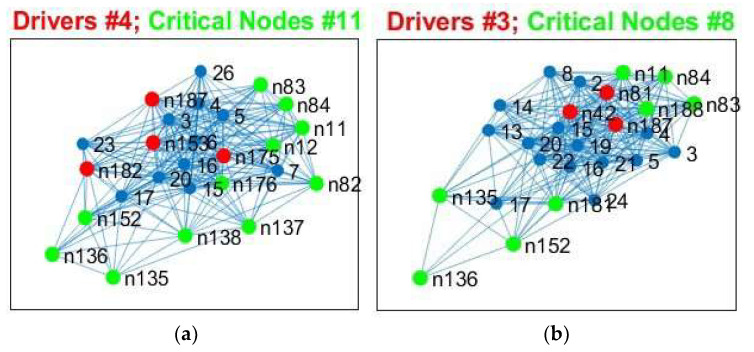
Critical Nodes for functional data, rsfMRI). (**a**) Controls, (**b**) IDHmut, and (**c**) IDHwt subjects. Critical nodes are in green and initial driver nodes are in red.

**Figure 10 cancers-15-02714-f010:**
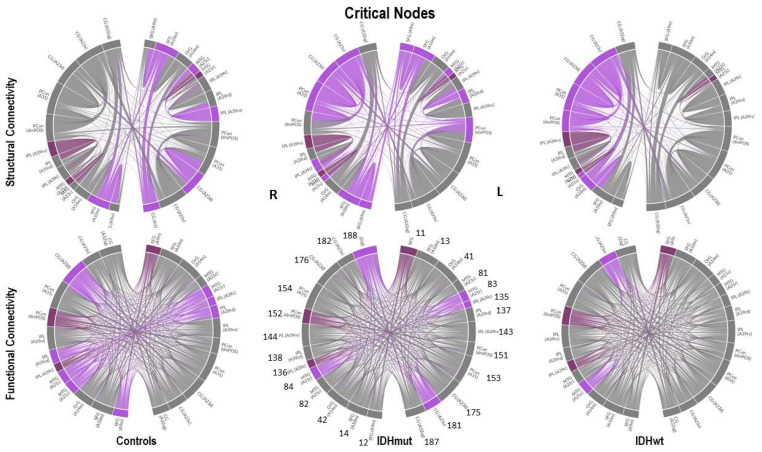
Connectograms for SC and FC for each group, highlighting in magenta the critical node (CN) connections. Given in dark magenta are common CN across groups, both for SC and FC, respectively. DMN nodes are labeled numerically according to the ROIs as given in the Brainnetome Atlas (https://atlas.brainnetome.org, accessed on 20 February 2020) The total number of CN for FC decreases in patients, with the lowest number in the IDHwt group. The total number of CN for SC remains stable in IDHwt and increases in IDHmut patients. Not only losses but also ”alternative” CN are found mainly in IDHmut patients compared to controls indicating a topological shift of CN in some of the patients.

**Table 1 cancers-15-02714-t001:** Clinical description of included patients.

Patients	IDH-Mutation	Diagnosis	Grade	Location	Side	Volume (in mm^3^)	Age (Years)	Education (Years) ***
1	Y	Oligodendroglioma *	II	Frontal	R	2	36–40	13
2	Y	Anaplastic astrocytoma	III	Frontal	L	21	50–55	13
3	Y	Oligodendroglioma *	II	Frontal	L	24	26–30	13
4	Y	Anaplastic astrocytoma	III	Parietal	R	25	56–60	9
5	Y	Astrocytoma	II	Frontal, insular	R	30	30–35	13
6	Y	Astrocytoma	II	Frontal, insular	L	32	30–35	15
7	Y	Anaplastic oligodendroglioma *	III	Frontal	L	39	50–55	15
8	Y	Astrocytoma	II	Frontal	L	51	26–30	16
9	Y	Astrocytoma	II	Frontal	L	54	26–30	13
10	Y	Astrocytoma	II	Parietal	L	73	56–60	18
11	Y	Anaplastic oligodendroglioma *	III	Frontal	R	96	30–35	18
12	Y	Anaplastic astrocytoma	III	Temporal, parietal	L	114	40–45	13
13	Y	Anaplastic astrocytoma	III	Parietal	L	119	20–25	13
14	Y	Anaplastic astrocytoma	III	Frontal	R	155	30–35	15
15	Y	Anaplastic oligodendroglioma *	III	Frontal	R	175	30–35	13
16	N	Glioblastoma multiforme	IV	Temporo-parietal	L	2	56–60	10
17	N	Glioblastoma multiforme **	IV	Frontal	L	11	76–80	13
18	N	Glioblastoma multiforme	IV	Temporo-parietal-occipital	L	11	60–65	15
19	N	Glioblastoma multiforme **	IV	Temporo-parietal	L	13	50–55	13
20	N	Glioblastoma multiforme	IV	Frontal	R	19	66–70	12
21	N	Glioblastoma multiforme	IV	Frontal, insular	L	20	66–70	9
22	N	Glioblastoma multiforme	IV	Temporal, parietal, occipital	L	25	66–70	12
23	N	Glioblastoma multiforme	IV	Occipital	L	44	50–55	13
24	N	Glioblastoma multiforme	IV	Parietal, occipital	R	47	76–80	9
25	N	Anaplastic astrocytoma	III	Frontal	L	49	40–45	13
26	N	Anaplastic astrocytoma	III	Temporal, parietal	L	51	70–75	18
27	N	Glioblastoma multiforme	IV	Parietal	L	64	60–65	9
28	N	Glioblastoma multiforme	IV	Temporal, parietal	L	116	66–70	18
29	N	Glioblastoma multiforme	IV	Frontal	R	121	56–60	16

Note. IDH = isocitrate-dehydrogenase, Y = yes, N = no, L = left, R = right. * Patients with codeletion of chromosome arms 1p and 19q. ** Recurrent tumor with preceding resection and adjuvant radiochemotherapy. *** Years of education were computed by the sum of years spent for school career and further training/study.

**Table 2 cancers-15-02714-t002:** Average number of driver nodes for controls (Con) and glioma cases (IDHmut, IDHwt) for structural and functional graph networks regarding the DMN.

DMN	Con	IDMmut	IDMwt
SC	9	7.8	5.6
FC	11.80	10.20	8.40

**Table 3 cancers-15-02714-t003:** Critical nodes for structural and functional networks. Abbreviations: SFG (Superior Frontal Gyrus), OrG (Orbital Gyrus), MTG (Middle Temporal Gyrus), IPL (Inferior Parietal Lobule), Pcun (Precuneus), and CG (cingulate Gyrus). Highlighted in red are nodes classified as critical nodes in patients but not in controls. Given in **bold** are critical nodes common across groups.

	Controls	IDHmut	IDHwt
Critical	13, 14, 81, 83, **84**	11, 12, 13, 14, 81, 82, 83, **84**	42, 82, 83, **84**, **144**
Nodes	143, **144**, 175, 187	136, 137, **144**, 151, 154, 176, 182	152, 154, 176, 182
SC	SFG	SFG	OrG
	MTG	MTG	MTG
	IPL	IPL	Pcun
	CG	CG	CG
Critical	**11**, 12, 82, 83, 84, 135,	**11**, 83, 84, 135, **136**, **152**,	**11**, 82, **136**, **152**, 182
Nodes	**136**, 137, 138, **152**, 176,	181, 188	-
FC	SFG	SFG	SFG
	MTG	MTG	MTG
	IPL	IPL	IPL
	Pcun	Pcun	Pcun
	CG	CG	CG

## Data Availability

The datasets are not publicly available due to data privacy protection obligations but are available from the corresponding author on reasonable request.

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
