# Peer review of "Controllability and Robustness of Functional and Structural Connectomic Networks in Glioma Patients"

_cancers, 2023, doi:10.3390/cancers15102714_

Round 1

Reviewer 1 Report (Previous Reviewer 3)

The authors have responded to reviewer comments very thoroughly and made this somewhat technical report more suitable for a broad audience.

Author Response

Reviewer 1 has not requested any changes. 

He said: The authors have responded to reviewer comments very thoroughly and made this somewhat technical report more suitable for a broad audience.

Reviewer 2 Report (Previous Reviewer 1)

The manuscript has in my opinion benefitted greatly from the performed revisions: the idea is clearer, results are presented more straightforwardly and a level of depth has been added. 

Issues remain with the following:

- Referring to a different paper for methods and putting the most basic scan parameters in supplement is still very unuseful in my opinion. A reader that has this paper in hand and wants to understand what was done, should be able to get the data and its collection. There is no word limit for Cancers, why not simply include such basic info in the current manuscript as well? It would save many people (imagine being on a plane or otherwise without wifi and not being able to download additional papers to understand the most basic efforts of this paper). 

- The inclusion of recurrent glioma patients, namely only n=2, perhaps impacts results: these patients will already have undergone tumor resection and may have very different drivers. Could the authors provide a replication of the findings without these 2? 

- Conceptually, I do find that the interpretation of differences between patients and controls to reflect tumor-induced shifts rather limited. There are more and more studies showing that intrinsic network features may also impact tumor growth and perhaps be preexisting/premorbid characteristics (e.g. Mandal et al 2020 Brain, Numan et al 2022 Brain). Please offer a more nuanced interpretation of your findings in the discussion. 

Author Response

We have attached a word-file with the answers and additional images.

Round 2

Reviewer 2 Report (Previous Reviewer 1)

Thanks for incorporating the suggestions. 

This manuscript is a resubmission of an earlier submission. The following is a list of the peer review reports and author responses from that submission.

Round 1

Reviewer 1 Report

This is a timely investigation, using the relatively new framework of network controllability in glioma. They use a moderate sample of glioma patients, which seem representative for the population, to assess structural and functional drivers of network control to compare with matched healthy controls. 

Major issue

- The section on controllability in the introduction does a suboptimal job of explaining exactly why and how the method is of use, and what it can and cannot do. What are signals in this context? How should we see a disease state? These things matter to set the stage, particularly in a general outlet with a wide audience. Likewise, the final paragraph of the introduction is all over the place, going from phenotyping to monitoring to detailed controllability wording. I think the authors should really stick with their actual research here: they did not do any monitoring, and should more specifically state their hypothesis on controllability in glioma. 

- Was 1p/19q codeletion tested? This is part of the WHO 2016 classification, but is never mentioned. 

- Although the controls are group-level matched to the glioma patients, I would prefer also splitting the control group for the IDH subgroup analyses. The IDHmut patients are so much younger and usually in more favorable condition than the IDHwt patients, that simply comparing with an average HC group could lead to misleading results. 

- The Data Acquisition section is too brief, particularly since I cannot see which 'previous studies' are referenced for detailed protocols. For both EPI sequences (which are used for the controllability analyses), basic sequence information, online and offline artefact processing etc should be in this manuscript (or at least supplementary). Moreover, I don't understand how tumor segmentation was used for DTI and rsfMRI, one would normally do the segmentation on an MPRAGE (and other anatomical scans) and then coregister the segmentation too the lowres EPI sequences. It is also unclear which Brainnetome regions were considered DMN. In conclusion, the whole method is very unclear as is. 

- Calculating the controllability only for the DMN is a major decision that seems somewhow hidden in a byline in the methods. This greatly impacts what the study does, how it compares to previous work on controllability, and potentially also the robustness of findings (since details on which nodes were DMN are not provided in methods). This choice should be clearly argumented for, I currently think this is a huge shortcoming of the work. 

- Section 2.4 is long and windy on the one hand, and extremely technical on the other hand. The section seems to conflate structural and funcitonal analyses, but of course, correlation is NOT used in structural network analysis. In fact, this section is mostly about thresholds, not about how connectivity was generated, which is extremely poorly explained only in lines 133-137. This is completely unreproducible as is. 

- There is no section on statistical methods used. 

- I read nowhere how a potential tumor IN the DMN (this happens quite often) is handled in the analysis, particularly since tumor tissue severely distorts EPI sequences. 

- I also do not read any indication of visual inspection of all the registrations between different template/native/functional/diffusion spaces, which is of high importance in brain tumor patients who almost inevitably have mass effects, edema, midline shift etc in their scans. 

- Part of the results contains new methods, for instance introducing the CCF. Moreover, it repeats some of the approaches that were actually in the methods already, such as the thresholding scheme. I am lost. 

- "The drivers have a range of 1-4 (mean=1.875; median=1)." Is this the number of drivers? Of how many DMN nodes considered? 

- "DMN/IDMmut/IDHwt classification showed no clear trend and the partial correlation method described in the literature promised better results." Unfathomable which analysis is referred to, and 'no clear trend' is also mystifying: was this by visual inspection? Are there any visuals for us readers to join this assessment? 

- All of the detailed analyses of driver/critical nodes are difficult to assess when so much of the basic processing and handling of tumor artifacts in the imaging is missing. Seeing as there is a huge lack of info, all of these results could represent nodes that happen to fall in different parts of the brain for different patients due to the different locations of the tumors. 

Minor comments

- "better controllability" (in the abstract) is a bit of a confusing term: previous work on control show that some areas of the brain typically have higher, some lower controllability. What is 'better' then? I would advise a less ambiguous and more factual term. 

- Terms are used interchangeably: is 'leaders' (line 85) the same as 'drivers'? I have never seen leaders as a control theoretical formulation, would advise not to confuse the reader with these varying terms. 

- Figure 1: The items on the x-axis are separate groups of subjects, so why draw a line through these data points? Also, error/SD bars are necessary. 

Reviewer 2 Report

Authors report a study on functional and structural connectivity in glioma patients, comparing their model with healthy volunteers. The manuscript is well written, even if it suffers many trooping and formatting errors that should be corrected to improve the readability. Strength and limitation are well described and commented. Some references cited in the text are missed (for example line 45 on page 1). There are some typing and formatting errors that must be corrected before pubblication

I suggest publication of the manuscript after minor revision.

Reviewer 3 Report

This manuscript describes new data addressing a relatively novel concept of network-level functionality in brain, which may influence symptoms and outcome in glioma patient. It provides some some interesting data in context of good and poorer prognosis glioma, which may differ in terms of remodelling of these functional networks which may have relevance in clinical decision making.

The content is interesting but it is not currently presented in a way that is suitable for a non-specialist readership.

Specific comments: 

Introduction

'Dynamical' used incorrectly for dynamic in several instances.

Line 63-64. No clear rationale is given for this statement re controlling networks for application of therapies. The relevance to glioma is also not well covered. Although it is covered in the discussion, a better explanation of potential clinical relevance should be included here.

Materials and Methods

A comment should be made on the fact that only patients with good KPS were included – is this a skewed data set?

There is also no information on stage of disease post op/biopsy post radiotherapy etc. It should at least be stated if all these patients are newly diagnosed.

I do not have sufficient expertise to comment in detail on the modelling approaches that are applied.

To me it is not clear if the authors come to a decision about which of 3 analyses methods are most robust and appropriate.

Results

Specifically in this section, the information is impenetrable for a non-expert readership and needs to be improved.

A flow chart of how the data were interpreted could help here.

In line 242-243, a description of the effect of IDHmut/wt needs to be clarified.

The statistical design and selection of N patients is not included. 

In my view it would be clearer to describe the normal situation first using each analysis method then compare with the disease states.

Section 3.3 line 339 is there a name or reference missing?

Line 373, the statement about counterproductive exclusion of information using CLA method needs additional justification.

Discussion

I would suggest including some sort of external validation eg from groups with other neurodegenerative diseases or trauma as part of the discussion if available.

The discussion of potential therapeutic relevance is nice here, especially the idea of relevance to prehabilitation.

Limitations

This section is nicely written and useful for the reader.